# Demographic and pathogen characteristics of incident bacterial meningitis in infants in South Africa: A cohort study

Yannick Nkiambi Kiakuvue[1], Sumaya Mall[1], Nelesh Govender[1,2,3,4,5,6], Anne von Gottberg[1,7], Rudzani Mashau[7], Susan Meiring[2], Cheryl Cohen[1,7] *

1 School of Public Health, Division of Epidemiology and biostatistics, University of the Witwatersrand, Johannesburg, South Africa, 2 Centre for Respiratory Diseases and Meningitis, National Institute for Communicable Diseases, Johannesburg, South Africa, 3 Division of Public Health Surveillance and Response, National Institute for Communicable Disease, a division of the National Health Laboratory Service, Johannesburg, South Africa, 4 Division of Medical Microbiology, Faculty of Health Sciences, University of Cape Town, Cape Town, South Africa, 5 MRC Centre for Medical Mycology, University of Exeter, Exeter, United Kingdom, 6 School of Pathology, Faculty of Health Sciences, University of Witwatersrand, Johannesburg, South Africa, 7 Centre for Healthcare-Associated Infections, Antimicrobial Resistance and Mycoses, National Institute for Communicable Diseases, Johannesburg, South Africa

* cherylc@nicd.ac.za

**Data Availability Statement:** Data are available from the National Health Laboratory Service Corporate Data Warehouse (NHLS-CDW)

## Abstract

### Introduction

Bacterial meningitis is a major cause of death, with an approximate case fatality rate of 37% across all age groups in South Africa. This study aimed to describe the demographic and pathogen characteristics of incident meningitis in children aged <1 year in South Africa from 2014 through 2018, during a period when *Haemophilus influenzae type b* vaccine and pneumococcal conjugate vaccines (PCV) were both included in the expanded program on immunization (EPI).

### Methods

We conducted a cohort study of routine laboratory data in the National Health Laboratory Service Corporate Data Warehouse, which covers approximately 80% of the South African population. We defined a case of laboratory-confirmed bacterial meningitis as any person aged <1 year with meningitis diagnosed by culture and identification of a pathogen documented as being a common cause of meningitis in CSF. The cause-specific incidence risks were calculated by dividing the number of positive specimens in each age group and year by the corresponding mid-year population for children under 1 year old and those in the post-neonatal period ($\geq$ 28 days to 365 days old). For children under 28 days old, the annual numbers of registered livebirths were used. We used Poisson regression to compare the incidence of meningitis by year.

### Results

We identified 3575 (1.5%) cases of culture-confirmed bacterial meningitis from the 232,016 cerebrospinal fluid (CSF) specimens tested from 2014–2018. The highest number of cases

Institutional Data Access for researchers who meet the criteria for access to confidential data.

**Funding:** Funding for the study was awarded as a grant to NG from the Bill & Melinda Gates Foundation (OPP1208882). The funders had no role in study design, data collection and analysis, decision to publish, or preparation of the manuscript.

**Competing interests:** The authors have declared that no competing interests exist.

were recorded in children aged <28 days (1873, 52.4%), male children (1800, 50.4%) as well as in the Gauteng Province (2014, 56.3%). *Acinetobacter baumannii* (14.9%), *followed* by *Klebsiella pneumoniae* (13.5%), *and* group B streptococcus (GBS) (10.7%), were the most common pathogens detected. Overall, *A. baumannii* had the highest incidence risk, occurring at 9.8 per 100,000 persons in children aged <1 year in 2018. Among neonates, *A. baumannii* peaked at 14.9 per 100,000 livebirths in 2018, while *Streptococcus pneumoniae* was most common in the post-neonatal period ($\geq$ 28 days to 365 days old), peaking at 9.8 per 100,000 persons in 2014. There was an increase in the annual incidence of most pathogens over the study period.

## Conclusion

There was an increasing trend in the annual incidence of bacterial meningitis in infants caused by most pathogens, particularly *A. baumannii*, *K. pneumoniae* and GBS. In addition to increased uptake of vaccination, prevention measures to reduce nosocomial and mother-to-child transmission of bacteria could include antenatal screening for GBS in pregnant women, rigorous hygiene in the hospital environment as well as rational antibiotic use.

## Introduction

Bacterial meningitis comes second in terms of frequency among the causes of meningitis in children, with viral meningitis being the most common [1, 2]. In Low-and Middle-Income Countries (LMIC), the bacterial meningitis fatality rate is estimated at 8% in children under 5 years old while it is estimated between 40 and 58% in neonates [3–6]. Global burden of disease data suggests that LMIC experience a higher burden of bacterial meningitis than high-income countries [7]. The consequences of meningitis pose serious implications for public health including mortality and long-term cognitive consequences [5, 8, 9].

Incidence estimates of meningitis vary globally, particularly between high-income and low-income countries. For example, in 2016, estimated overall bacterial meningitis incidence varied from 0.5 per 100 000 persons in Australia to 207.4 per 100 000 persons in South Sudan. The variations observed could arise from true differences in disease burden and, to some extent, from factors such as lack of consistency in surveillance systems of different regions leading to the underreporting of cases [7, 10]. However, in 2016, 2.8 million cases of bacterial meningitis were estimated in all ages in the world and 56% of them were among children under 5 years with an estimated annual incidence of 143.6 per 100 000 persons [11]. In South Africa, from 2014 to 2018, Mashau and colleagues reported an overall incidence risk of neonatal bloodstream infection and meningitis estimated at 6.0 per 1000 livebirths [12].

There is also variation in aetiology (pathogen characteristics) of bacterial meningitis and aetiology may change over time with the introduction of new vaccines [10]. Globally, bacterial meningitis aetiology varies by geographic region and age group. *S. pneumoniae* and *Neisseria meningitidis* are still the more predominant pathogens in all age groups and all regions despite the introduction of their respective vaccines, whereas the incidence of *Haemophilus influenzae type b* has declined since the introduction of the Hib vaccine [7, 13]. *Escherichia coli*, GBS, and *Listeria monocytogenes* are the most frequent bacteria that cause meningitis in the neonatal period around the world [14]. *N. meningitidis* is known to be a major cause of endemic and epidemic meningitis around the world [10]. A study conducted from 2014 to 2019 among

South African neonates aged <28 days, using a similar dataset to the one we have used, found that *A. baumannii*, *K. pneumoniae*, GBS were the most common pathogens detected in the CSF specimens [12].

Several meningitis vaccines are included in routine immunisation schedules or are used to control outbreaks around the world [15]. In South Africa, the PCV-7 vaccine was introduced in 2009 replaced by PCV-13 in 2011 [15–17]. The Hib vaccine was introduced in 1999 as part of the national immunisation programme and the Hib vaccine booster dose in 2010 [18, 19]. A quadrivalent protein-polysaccharide meningococcal vaccine is available in South Africa but is not part of the public South African immunisation programme [15, 19, 20].

To date, there are a paucity of nationally representative data assessing the incidence and associated characteristics (including the pathogens described above) of bacterial meningitis (excluding tuberculosis (TB)) after introduction of different vaccines. The purpose of this study was to describe the incidence risk and aetiology of bacterial meningitis in children aged <1-year-old in South Africa from year 2014 through 2018 while Hib and PCV-13 vaccines were in use.

## Materials and methods

### Study population and study design

We conducted a retrospective cohort analysis of routine laboratory data. Our study population included any person aged <1 year who underwent a lumbar puncture at a public sector facility in South Africa and had a cerebrospinal fluid (CSF) specimen sent to an NHLS laboratory. We specifically focused on cases where a bacterial organism, considered a pathogen causing meningitis, was identified through culture between January 2014 and December 2018.

### Source of data

Our data were extracted from the National Health Laboratory Service (NHLS) Corporate Data Warehouse (CDW). The NHLS is South Africa's largest diagnostic pathology service, serving public-sector facilities in all nine provinces. It provides laboratory diagnostics for an estimated 80% of the population of South Africa [21]. All routine laboratory results with accompanying demographic data from NHLS laboratory-processed specimens are electronically archived in the CDW [22]. Data were extracted on all CSF specimens submitted to NHLS laboratories from children aged <1 year from 2014 to 2018. Mid-year population and livebirths data from the Statistics South Africa (Stats SA) were used for the incidence calculations [23]. Our data consisted of pathology records collected from 210 healthcare facilities.

### Definitions

We defined a case of laboratory-confirmed bacterial meningitis as any person aged <1 year with meningitis diagnosed by culture and identification of a pathogen documented as being a common cause of meningitis from CSF including *N. meningitidis*, *H. influenzae*, *K. pneumoniae*, Group B *Streptococcus* (GBS), *Klebsiella* spp., *S. pneumoniae*, *E. coli*, *L. monocytogenes*, *A. baumannii*, *Serratia marcescens*, *Enterobacter cloacae*, *Klebsiella oxytoca*, *Pseudomonas aeruginosa*, *Streptococcus anginosus* etc. Any specimen sample with an organism that is not documented as being a common cause of meningitis and which can commonly contaminate CSF sample (e.g.: *Aerococcus viridans*, *Bacillus cereus*, *Bacillus spp*, *Corynebacterium* species; *Staphylococcus capitis*, *Staphylococcus cohnii*, *Staphylococcus epidermidis*, *Staphylococcus haemolyticus*, *Staphylococcus hominis*, *Staphylococcus intermedius*, *Staphylococcus lentus*, *Staphylococcus saprophyticus*, *Staphylococcus urealyticus*, *Staphylococcus warneri*, *Streptococcus gordonii*,

*Streptococcus mitis*, *Streptococcus oralis*, *Streptococcus parasanguinis*, *Streptococcus sanguinis*, *Streptococcus salivarius*, *Streptococcus vestibularis*,) was defined a contaminant. As recommended by the US Centers for Disease Control and Prevention (CDC), a case of coagulase-negative *Staphylococcus* (CoNS) infection was defined as two or more separate cultures isolated within two days of the original culture [24]. CoNS was considered a contaminant in each case where it was isolated from only one specimen or from two specimens, but the second specimen was collected more than 48 hours later. Cases of TB were not included in the study.

As individuals could have had multiple CSF specimens submitted during the study period, we defined an episode of illness as all positive samples within a 14-day period. If a sample tested positive more than 14 days after a previous episode, it was considered a new episode.

The post-neonatal period was defined as the interval from $\geq$28 to 365 days of age, while the neonatal period encompassed the time frame from birth to <28 days of age.

The seasons were defined accordingly to the South African meteorological reports as followed: Summer: December, January, February; Autumn: March, April, May; Winter: June, July, August; Spring: September, October, November [25].

Only cases with a recorded date of birth were included. For cases with missing date of birth, but with ward name marked as kangaroo mother care (KMC) unit, neonatal/nursery were included in the analysis, and the ward name was used a proxy for being neonates.

## Statistical analysis

A number of steps were undertaken in the analysis phase:

First, a descriptive analysis of the aetiology of bacterial meningitis was conducted using frequency (n) and percentage (%) tables. We calculated proportions by dividing the number of cause-specific cases by the total number of cases of bacterial meningitis. Only the six most commonly detected pathogens were included in this part of the study, other pathogens were grouped under "other" pathogen category.

Secondly, the incidence risk of specific pathogens were calculated by dividing the age-group, year of the specimen and the number of positive specimens by the appropriate mid-year population. Livebirths were used for the calculation of incidence risk among children under 28 days old, while the appropriate under-1-year population was estimated from the Stats SA Sprague table [26]. For children $\geq$ 28 days to 365 days old, the population was estimated by subtracting the livebirths from the under-1-year appropriate population. Given that people who use public facilities account for 80% of the population, the mid-year population was then multiplied by 0.8 [21]. All incidence risks among children aged <28 days were expressed per 100 000 livebirths and per 100 000 persons for the two other age groups (under 1 year and $\geq$ 28 days to 365 days old). Data on the Western Cape Province for the year 2014 was not included in any incidence calculations because of the missing CSF data for the same year. Poisson regression was used to determine if the incidence risk trends were different by year. Only the six most commonly detected pathogens were included in the distribution calculations, while pathogens with fewer cases were grouped under "other" pathogen. Stata (Statacorp, College Station, Texas USA), version 15 was used for data management and all statistical analyses.

## Ethics statement

This study was approved by the University of the Witwatersrand Human Research Ethics (Medical) Committee (M210110). All data were protected and accessed on 30/04/2021 after ethical approval. Co-authors did not have access to individual participants and could not identify them. Additionally, all data were fully anonymized before being accessed, ensuring the confidentiality and privacy of the participants.

## Results and discussion

### Results

During the five-year study period (2014–2018), a total of 3575 (1.5%) cases of culture-confirmed bacterial meningitis were identified from the 232,016 CSF specimens tested by the NHLS in South Africa (Fig 1). Of the 3575 cases, 52.4% (n = 1873) were in neonates aged <28 days, 50.4% (n = 1800) were males, 56.3% (n = 2014) were recorded in Gauteng Province, and 26.9% (n = 963) occurred in the Spring (Table 1).

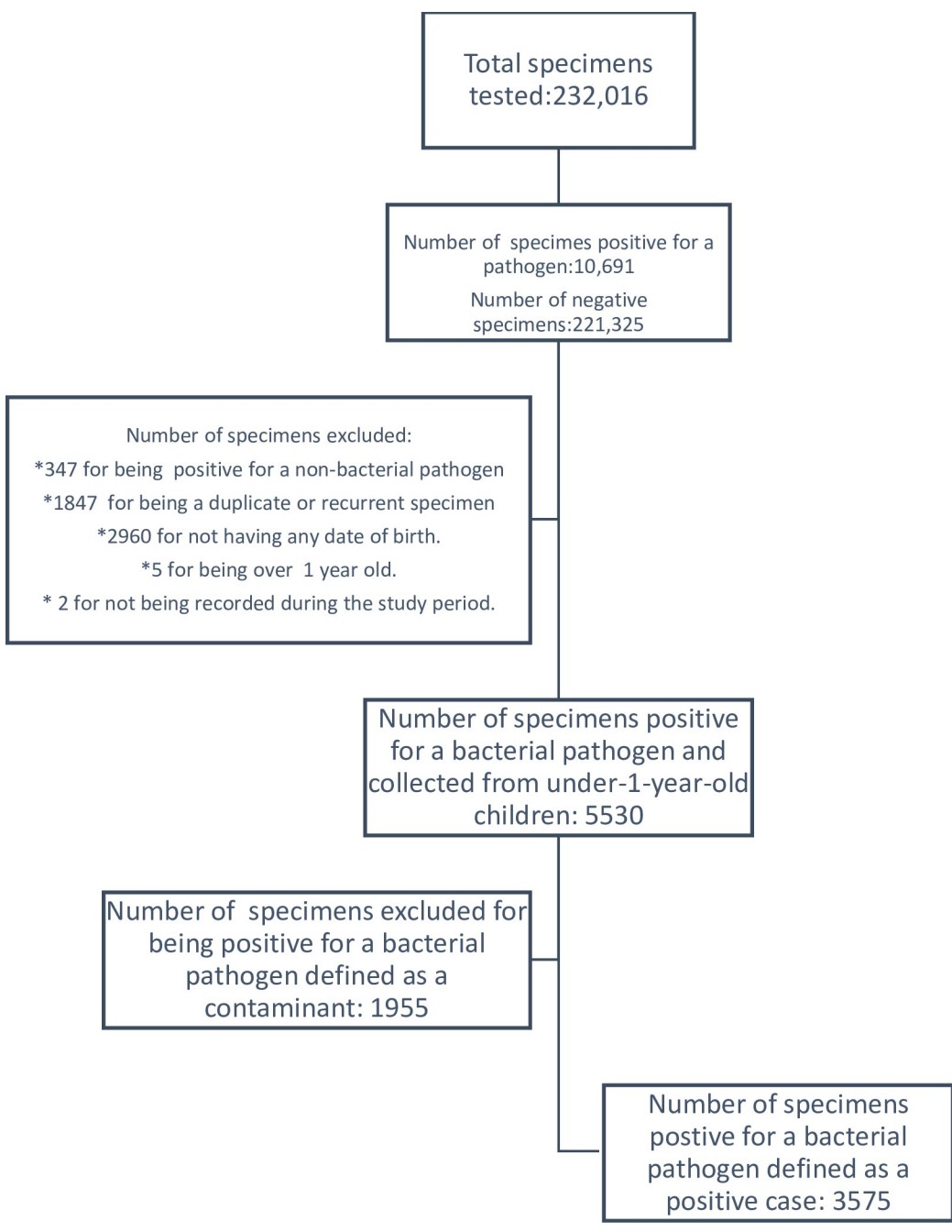

**Fig 1. Flow diagram of the population included in the study.**

**Table 1. Demographic characteristics of patients with laboratory-confirmed meningitis infection due to bacterial pathogen in South Africa, 2014–2018.**

| Pathogens | A. baumannii | K. pneumoniae | GBS | S. pneumoniae | S. aureus | E. faecium | Others | TOTAL |
|---|---|---|---|---|---|---|---|---|
| | n (%) | n (%) | n (%) | n (%) | n (%) | n (%) | n (%) | n (%) |
| **TOTAL** (n/N) | 536(14.9) | 484(13.5) | 384(10.7) | 372(10.4) | 321(8.9) | 224(6.3) | 1,308(32.2) | 3575(100) |
| **AGE-GROUP** (days) | | | | | | | | |
| <28 | 388(72.4) | 304(62.8) | 287(74.7) | 60(16.1) | 120(37.4) | 137(61.2) | 578(46.1) | 1873(52.4) |
| ≥28 | 148(27.6) | 180(37.2) | 97(25.3) | 312(83.9) | 201(62.6) | 87(38.8) | 677(53.9) | 1702(47.6) |
| **SEX** | | | | | | | | |
| Female | 223(41.6) | 213(44) | 188(48.9) | 172(46.2) | 144(44.9) | 102(45.5) | 540(43) | 1582(44.2) |
| Male | 272(50.7) | 242(50) | 178(46.4) | 183(49.2) | 162(50.5) | 111(49.6) | 653(52) | 1800(50.4) |
| Unknown | 41(7.7) | 29(6) | 18(4.7) | 17(4.6) | 15(4.6) | 11(4.9) | 62(5) | 193(5.4) |
| **PROVINCE** | | | | | | | | |
| Eastern cape | 34(6.3) | 44(9.1) | 36(9.4) | 41(11) | 18(5.6) | 6(2.7) | 96(7.6) | 275(7.7) |
| Free state | 28(5.2) | 12(2.5) | 12(3.1) | 27(7.3) | 11(3.4) | 8(3.6) | 47(3.7) | 145(4.1) |
| Gauteng | 374(69.8) | 288(59.5) | 197(51.3) | 164(44.1) | 195(60.8) | 167(74.6) | 630(50.2) | 2014(56.3) |
| KwaZulu-Natal | 40(7.5) | 71(14.7) | 52(13.5) | 43(11.6) | 47(14.6) | 25(11.2) | 197(15.7) | 475(13.3) |
| Limpopo | 25(4.7) | 23(4.8) | 13(3.4) | 19(5.1) | 13(4.1) | 5(2.2) | 78(6.2) | 176(4.9) |
| Mpumalanga | 12(2.2) | 9(1.9) | 13(3.4) | 13(3.5) | 2(0.6) | 3(1.3) | 29(2.3) | 81(2.3) |
| North West | 2(0.4) | 8(1.7) | 8(2.1) | 25(6.7) | 1(0.3) | 0(0) | 32(2.5) | 76(2.1) |
| Northern cape | 4(0.7) | 5(1) | 3(0.8) | 5(1.3) | 4(1.3) | 4(1.8) | 20(1.6) | 45(1.2) |
| Western cape | 17(3.2) | 24(4.9) | 50(13) | 35(9.4) | 30(9.4) | 6(2.7) | 126(10) | 288(8.1) |
| **YEAR** | | | | | | | | |
| 2014 | 52(9.7) | 67(13.8) | 63(16.4) | 66(17.7) | 55(17.1) | 24(10.7) | 184(14.6) | 511(14.3) |
| 2015 | 85(15.9) | 75(15.5) | 78(20.3) | 88(23.7) | 71(22.1) | 48(21.4) | 214(17.1) | 659(18.4) |
| 2016 | 85(15.9) | 99(20.45) | 65(16.9) | 70(18.8) | 56(17.5) | 48(21.4) | 317(25.3) | 740(20.7) |
| 2017 | 151(28.2) | 110(22.7) | 87(22.7) | 74(19.9) | 58(18.1) | 35(15.6) | 262(20.9) | 777(21.7) |
| 2018 | 163(30.4) | 133(27.48) | 91(23.7) | 74(19.9) | 81(25.2) | 69(30.8) | 278(22.2) | 888(24.8) |
| **MONTH** | | | | | | | | |
| January | 40(7.5) | 50(10.3) | 31(8) | 21(5.7) | 28(8.7) | 12(5.4) | 94(7.5) | 276(7.7) |
| February | 44(8.2) | 45(9.3) | 32(8.3) | 25(6.7) | 23(7.1) | 10(4.5) | 108(8.6) | 287(8) |
| March | 38(7.1) | 53(10.9) | 26(6.8) | 30(8.1) | 35(10.9) | 22(9.8) | 111(8.8) | 315(8.8) |
| April | 45(8.4) | 37(7.6) | 29(7.5) | 35(9.4) | 40(12.5) | 12(5.4) | 105(8.4) | 303(8.5) |
| May | 47(8.8) | 37(7.6) | 34(8.8) | 35(9.4) | 38(11.8) | 23(10.3) | 117(9.3) | 331(9.3) |
| June | 40(7.5) | 29(5.9) | 31(8) | 42(11.3) | 22(6.8) | 23(10.3) | 98(7.8) | 285(7.9) |
| July | 35(6.5) | 40(8.3) | 28(7.3) | 36(9.6) | 21(6.5) | 21(9.4) | 97(7.7) | 278(7.8) |
| August | 48(8.9) | 36(7.4) | 42(10.9) | 30(8.1) | 23(7.20) | 21(9.4) | 89(7.1) | 289(8.1) |
| September | 43(8) | 31(6.4) | 34(8.8) | 40(10.7) | 19(5.9) | 24(10.7) | 120(9.6) | 311(8.7) |
| October | 42(7.8) | 36(7.4) | 38(9.9) | 30(8.1) | 27(8.4) | 26(11.6) | 115(9.2) | 313(8.8) |
| November | 79(14.7) | 47(9.7) | 34(8.8) | 30(8.1) | 26(8.1) | 16(7.1) | 107(8.5) | 339(9.5) |
| December | 35(6.5) | 43(8.9) | 25(6.5) | 18(4.8) | 19(5.9) | 14(6.2) | 94(7.5) | 248(6.9) |
| **SEASON** | | | | | | | | |
| Summer | 119(22.2) | 138(28.5) | 88(22.9) | 64(17.2) | 70(21.8) | 36(16.1) | 296(23.6) | 811(22.7) |
| Autumn | 130(24.2) | 127(26.3) | 89(23.2) | 100(26.9) | 113(35.2) | 57(25.5) | 333(26.5) | 949(26.5) |
| Winter | 123(22.9) | 105(21.7) | 101(26.3) | 108(29) | 66(20.6) | 65(29) | 284(22.6) | 852(23.8) |
| Spring | 164(30.6) | 114(23.6) | 106(27.6) | 100(26.9) | 72(22.4) | 66(29.5) | 342(27.2) | 963(26.9) |

n: frequency; %: column percentage; GBS–Group B streptococcus

## Aetiology

*A. baumannii* was the most common pathogen identified (536, 14.9%) followed by *K. pneumoniae* (484, 13.5%), GBS (384, 10.7%), *S. pneumoniae* (372, 10.4%), *S aureus* (321, 8.9%), *E. faecium* (224, 6.3%) (Table 1). *E. faecalis. E. coli*, *N. meningitidis*, *P. aeruginosa*, *H. influenzae*, *E. cloacae*, *S. marcescens*, and *S. maltophilia* were among the other pathogens detected.

The most common pathogens detected among neonates aged <28 days were *A. baumannii* (388, 20.7%), *K. pneumoniae* (304, 16.2%), and GBS (287, 15.3%), while the most common pathogens detected among those in post-neonatal age group were *S. pneumoniae* (312, 18.3%), *S. aureus* (201, 11.8%), and *K. pneumoniae* (180, 10.6%). More than 70% of *A. baumannii* cases were observed in neonates, while more than 80% of cases of *S. pneumoniae* were recorded in children in the post-neonatal period. *A. baumannii* was the most common pathogen identified in both male and female children (Table 1).

When examined by province, *A. baumannii* was as the most common pathogen in Gauteng (374, 18.6%), Limpopo (25, 14.2%) and the Free State provinces (28, 19.3%), while *K. pneumoniae* dominated in the Eastern Cape (44, 16%) and KwaZulu-Natal provinces (71, 14.9%). In Mpumalanga (13, 16%) and the Western Cape provinces (50, 17.3%), GBS was the most frequently recorded pathogen, while in the North West Province, *S. pneumoniae* (25, 32.9%) was most common. *A. baumannii* was the most common pathogen detected in the Spring (164, 17%), Winter (123, 14.4%), and the Autumn (130, 13.7%), while *K. pneumoniae* was the most common in the Summer (138, 17%) (Table 1).

In the overall sample, *N. meningitidis* and *H. influenzae* were relatively rare, accounting for 1.4% (51 cases) and 2.2% (79 cases) of infections, respectively, and were thus categorized under other pathogens. Among the identified cases, 80.4% (41 cases) of *N. meningitidis* and 92.4% (73 cases) of *H. influenzae* cases were recorded in children aged ≥28 days.

## Incidence

The annual incidence risk of most pathogens increased each year in all ages categories (all under 1 year, neonate, post-neonate) (Figs 2–4). The annual incidence risk among children

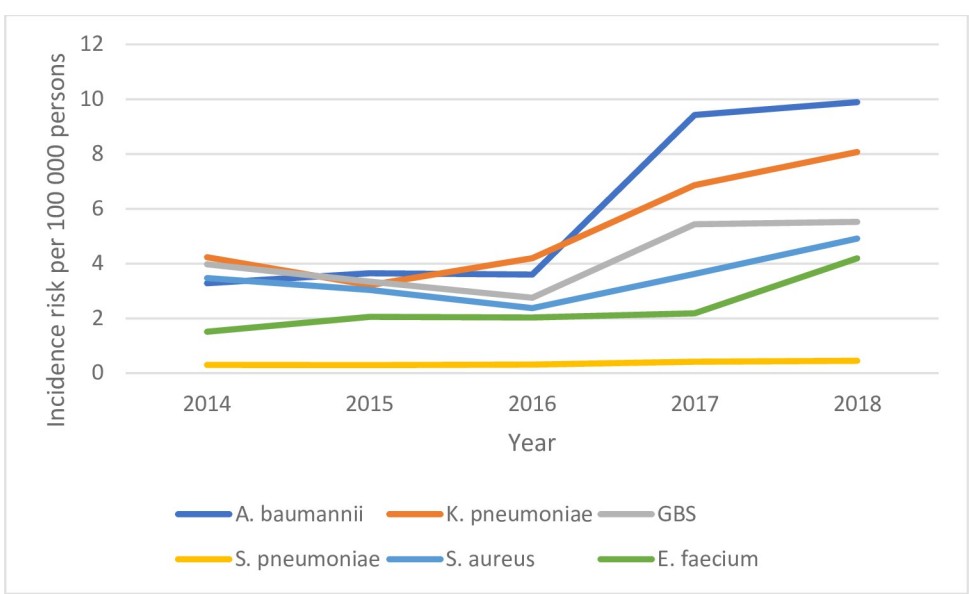

**Fig 2. Incidence risk of the most common pathogens among children under 1-year-old by year, South Africa, 2014–2018.**

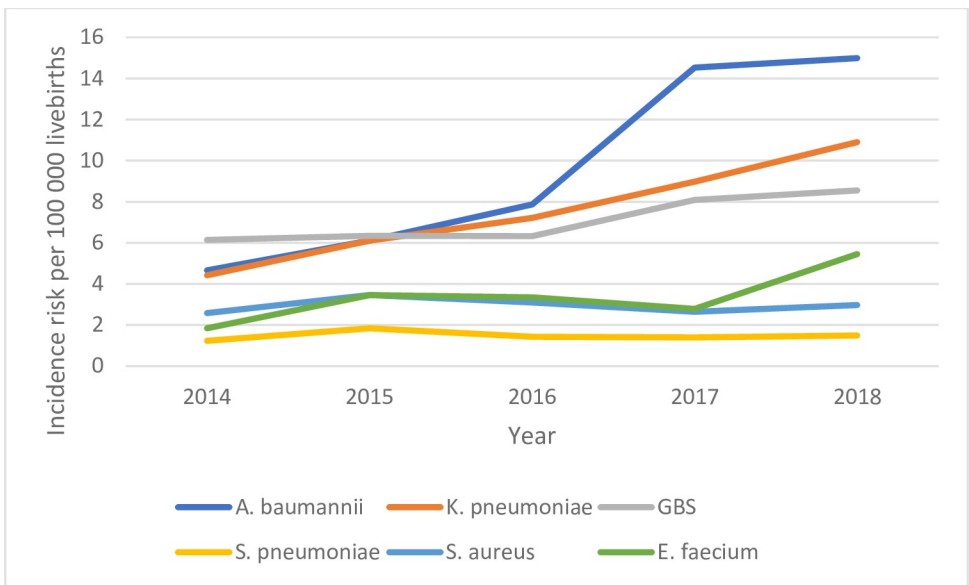

**Fig 3. Incidence risk of the most common pathogens detected among children <28 days old, South Africa, 2014–2018.**

aged <1 year varied by pathogen and year. With an incidence risk of 9.8 per 100 000 persons (95% CI, 8.4–11.5) in children under 1 year old in 2018 and 14.9 per 100 000 livebirths (95% CI 12.4–17.9) in children under 28 days old in 2018, *A. baumannii* had the highest incidence risk calculated over the study period in these two age groups. In children ≥28 days (post-neonatal period), the highest incidence risk was for *S. pneumoniae* (9.8 per 100 000 persons (95% CI 7.3–12.7)) (Table 2).

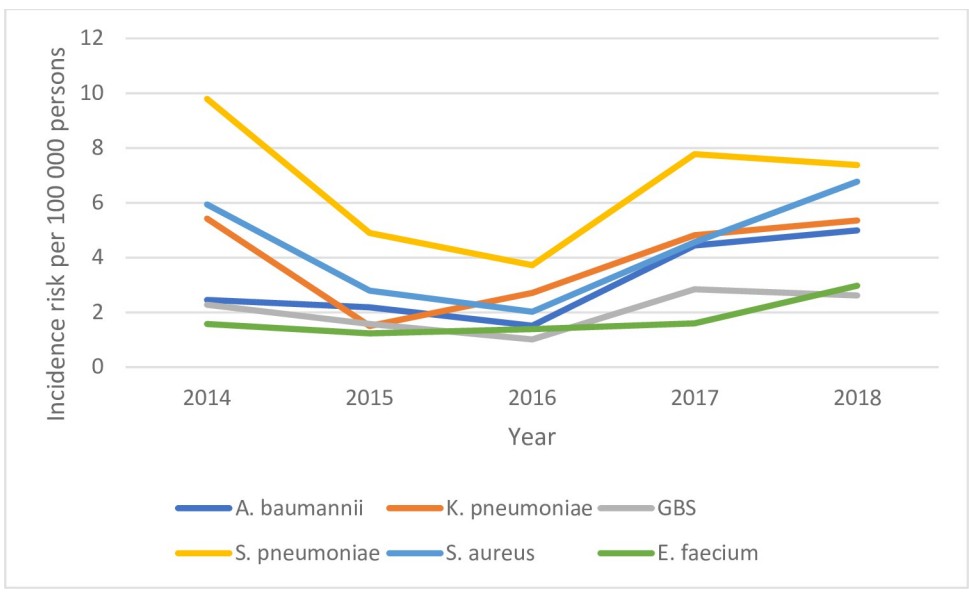

**Fig 4. Incidence risk of the most common pathogens detected among children ≥28 days old, South Africa, 2014–2018.**

**Table 2. Incidence risk (95% CI) per 100 000 persons of the most common pathogens detected among children under 1 year old, South Africa, 2014–2018.**

| Age group | 2014 | 2015 | 2016 | 2017 | 2018 |
|---|---|---|---|---|---|
| **All** | | | | | |
| *A. baumannii* | 3.3 (2.5–4.3) | 3.6 (2.9–4.5) | 3.6 (2.9–4.5) | 9.4 (7.9–11.1) | 9.8 (8.4–11.5) |
| *K. pneumoniae* | 4.2 (3.3–5.4) | 3.2 (2.5–4.0) | 4.2 (3.4–5.1) | 6.9 (5.6–8.3) | 8.1 (6.8–9.6) |
| *S. pneumoniae* | 4.2 (3.2–5.3) | 3.8 (3.0–4.6) | 2.9 (3.5–4.6) | 4.6 (3.6–5.8) | 4.5 (3.5–5.2) |
| GBS | 3.9 (2.6–4.2) | 3.3 (2.6–4.2) | 2.7 (2.1–3.5) | 5.4 (4.3–6.7) | 5.5 (4.4–6.7) |
| *S. aureus* | 3.5 (2.6–4.5) | 3.0 (2.3–3.8) | 2.4 (1.8–3.1) | 3.6 (2.7–4.6) | 4.9 (3.9–6.1) |
| *E. faecium* | 1.5 (0.9–2.3) | 2.1 (1.5–2.7) | 2.0 (1.5–2.7) | 2.2 (1.5–2.7) | 4.2 (3.3–5.3) |
| **<28 days old** | | | | | |
| *A. baumannii* | 4.2 (2.9–5.7) | 6.1 (4.6–7.9) | 7.9 (6–10.1) | 14.5 (12–17.4) | 14.9 (12.4–17.9) |
| *K. pneumoniae* | 3.9 (2.8–5.5) | 6.1 (4.6–7.9) | 7.2 (5.5–9.4) | 8.9 (7.0–11.3) | 10.9 (8.7–13.4) |
| *S. pneumoniae* | 1.1 (0.5–2.0) | 1.8 (1.1–2.9) | 1.4 (0.7–2.5) | 1.3 (0.6–2.4) | 1.5 (0.8–2.6) |
| GBS | 5.5 (4.1–7.2) | 6.3 (4.8–8.3) | 6.3 (4.7–8.4) | 8.1 (6.2–10.3) | 8.5 (6.7–10.8) |
| *S. aureus* | 2.3 (1.4–3.5) | 3.5 (2.3–4.9) | 3.1 (1.9–4.9) | 2.7 (1.6–4.1) | 2.9 (1.9–4.4) |
| *E. faecium* | 1.6 (0.9–2.7) | 3.5 (2.3–4.9) | 3.4 (2.2–4.9) | 2.8 (1.7–4.9) | 5.5 (3.9–7.3) |
| **≥28 to 365 days old** | | | | | |
| *A. baumannii* | 2.1 (1.1–3.5) | 2.2 (1.5–3.1) | 1.5 (0.9–2.3) | 4.4 (3.1–6.14) | 4.9 (3.6–6.8) |
| *K. pneumoniae* | 4.6 (3.1–6.6) | 1.5 (0.9–2.3) | 2.7 (1.9–3.7) | 4.8 (3.4–6.6) | 5.4 (3.9–7.2) |
| *S. pneumoniae* | 8.3 (6.3–10.8) | 4.9 (3.8–6.2) | 3.7 (2.8–4.8) | 7.8 (5.9–9.9) | 7.4 (5.7–9.5) |
| GBS | 1.9 (1.0–3.3) | 1.6 (0.9–2.4) | 1.0 (0.6–1.6) | 2.8 (1.8–4.3) | 2.6 (1.6–3.9) |
| *S. aureus* | 5.1 (3.5–7.1) | 2.8 (2.0–3.8) | 2.0 (1.4–2.8) | 4.6 (3.2–6.3) | 6.8 (5.1–8.8) |
| *E. faecium* | 1.3 (0.6–2.6) | 1.2 (0.7–1.9) | 1.4 (0.8–2.1) | 1.6 (0.8–2.7) | 2.9 (1.9–4.4) |

GBS: group B streptococcus

Almost all of the most common pathogens detected in different provinces increased over time, with the exception of *S. aureus*, *S. pneumoniae* and GBS, which increased from 2014 to 2017 before slightly decreasing in most provinces in 2018. Gauteng province had the highest incidence risk for all pathogens.

## Discussion

We described a 5-year incidence of culture-confirmed bacterial meningitis among infants in South Africa, bringing to light several important findings. There was an increase in the annual incidence of most pathogens over the study period. *A. baumannii*, followed by *K. pneumoniae* and GBS were the most common pathogens detected throughout the study. *A. baumannii*, *K. pneumoniae*, and GBS were most frequently identified in neonates, while in the post-neonatal period, *S. pneumoniae*, *S. aureus*, and *K. pneumoniae* were the most common pathogens. Finally, the Gauteng Province recorded the highest proportions and incidences of all pathogens.

We found an increasing (annual) incidence of most pathogens during the study period, a finding that is not always corroborated in previous studies. Several studies have found a decrease in the incidence of bacterial meningitis and its causative pathogens worldwide as well as in South Africa (in adults) following the introduction of various vaccines [27–32]. Our findings are consistent with what Mashau and colleagues discovered in children under the age of 28 days using the same dataset but describing cases of neonatal meningitis or bloodstream infection; they also discovered an increasing trend in the overall incidence risk of neonatal meningitis [12].

South Africa has established a long-standing vaccination program targeting pneumococcus and Hib [33]. However, there were some cases of *S. pneumoniae* as well as *H. influenzae*; the increasing trend cannot be attributed to these organisms. The rise in the incidences of bacterial meningitis can be attributed to several factors: increased bacterial transmission, especially among premature neonates; the prevalence of antibiotic-resistant strains; and changes in healthcare practices. Social factors like poverty and overcrowding could also exacerbate the spread of meningitis. Additionally, outbreaks in specific facilities, advancements in laboratory techniques, and improved NHLS CDW data collection and treatment contribute to this trend.

We found that *A. baumannii*, *K. pneumoniae*, and GBS were the most common pathogens detected during the study period. The current study's findings indicate a high proportion and predominance of pathogens such as *A. baumannii*, *K. pneumoniae*, *S. aureus* that can be classified as healthcare-associated infections and vertical transmission mostly for GBS. Inadequate infection control programmes (poor hand hygiene, poor environmental cleaning, disinfection and sterilization, poor antibiotic stewardship) prolonged hospital stays, a large number of premature births as well as a lack of antenatal screening and education for GBS in pregnant women could explain these trends [34–37].

The leading causes of bacterial meningitis in our study differed from what previous studies, also from LMICs, had suggested. In one study conducted in the Gulf region of Oman, for example, the introduction of meningococcal serogroup A and C vaccine, pneumococcal vaccine, and Hib vaccine had no effect on the leading pathogens. *H. influenzae S. pneumoniae*, and *N. meningitidis* remained the top three causes of bacterial meningitis in children under 1 year [38]. That could be because the Oman study used neutrophilic pleocytosis, a latex agglutination test for antigen, and a CSF culture test to define a case of bacterial meningitis, whereas our study only included the CSF culture test. Additionally, our study had a significantly higher number of neonatal infections compared to the study in Oman. In neonates, our fundings were comparable to what Mashau and colleagues found using a similar dataset, but the proportions of different pathogens were not the same [12]. In the post-neonatal period, the leading causative pathogens of bacterial meningitis in our study were different from what was found in Namibia, where *S. pneumoniae*, *Haemophilus spp*, and *Streptococcus spp* were the three most common pathogens isolated among children 1 month to 11 months of age, even though the first pathogen was the same [39]. This could be due to differences in the healthcare systems of the two countries as well as the study period (2009–2012).

The fact that Gauteng Province reported most cases is understandable given that it is the most populated province, accounting for 25.3% of the national population in 2018, as well as being highly urbanised, with better access to healthcare and laboratory testing [21, 40].

## Strengths and limitations

The research has several strengths. The study included all bacterial pathogens (other than tuberculosis) and data from all over South Africa. Despite the fact that the study only included CSF culture, it provided a clear distribution of aetiology and incidence of bacterial meningitis over a 5-year period using data from South Africa's largest pathology diagnostic service with good facilities for microbiological testing and procedures, which serves 80% of the total population [41].

We have also identified several limitations in our study. Firstly, we only considered cases where a laboratory examination was requested and performed. In some instances, a clinical diagnosis and treatment may have been instituted without a laboratory examination. Additionally, some cases may have been CSF culture-negative not because of the absence of a pathogen but because effective treatment was administered before the laboratory examination,

leading to false negatives. These factors could result in an underestimation of the incidence and a distortion of the actual aetiology.

Secondly, the incidence risks of specific pathogens were calculated by dividing the number of positive specimens by the appropriate mid-year population, categorized by age group and year of specimen collection. For children under 28 days old, livebirths were used for the calculation of incidence risk, while for the under-1-year population, estimates were derived from the Stats SA Sprague table. This method of estimation adds another layer of limitation, as it may not fully capture the precise population at risk.

Additionally, children using public sector services might differ in various sociodemographic and health-related aspects compared to those using private sector services. This difference might influence our incidence estimates and should be considered when interpreting the results.

Moreover, the absence of gestational age data for mothers and the lack of information on preterm surviving children made it difficult to accurately identify premature neonates. This limitation hindered our ability to precisely attribute the high incidence rate of bacterial meningitis to this particular population group. We also had missing data for the entire Western Cape Province in 2014 and missing dates of birth for 2960 specimens. Furthermore, we were unable to classify community-acquired versus nosocomial-acquired infections due to the unavailability of admission dates.

## Conclusion

The incidence of various pathogens increased during our study period. *A. baumannii*, *K. pneumoniae*, and GBS were the most common causative pathogens in neonates and children under one year old, while in the post-neonatal period, *S. pneumoniae*, *S aureus*, *K. pneumoniae* were the most common pathogens identified. Overall, this study suggests that nosocomial pathogens and pathogens transmitted from mother to child are the most common causes of bacterial meningitis, and that *S. pneumoniae* is the most common pathogen recorded in children in the postnatal period. Different public health programs should not only focus on vaccination (immunization) programs. National and provincial infection prevention programmes, antenatal screening for GBS in pregnant women and hygienic practice in the labour room and intensive care units, as well as a rationalization of antibiotic use to prevent nosocomial and mother to child transmissible pathogens should also be considered. In addition, further studies could collect clinical variables for a more in-depth characterisation of bacterial meningitis.

## Acknowledgments

We thank the National Health Laboratory Service (NHLS) Corporate Data Warehouse (CDW) for allowing us to use their data. We also thank the Centre for Respiratory Diseases and Meningitis team of National Institute for Communicable Disease (NICD) for their assistance.

## Author Contributions

**Conceptualization:** Nelesh Govender, Anne von Gottberg, Rudzani Mashau, Susan Meiring, Cheryl Cohen.

**Formal analysis:** Yannick Nkiambi Kiakuvue.

**Methodology:** Yannick Nkiambi Kiakuvue.

**Supervision:** Sumaya Mall, Cheryl Cohen.

**Visualization:** Yannick Nkiambi Kiakuvue.

**Writing – original draft:** Yannick Nkiambi Kiakuvue.

**Writing – review & editing:** Sumaya Mall.

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
