## [Decision Letter · Decision Letter 0]

8 Apr 2024

PONE-D-24-04693Demographic and pathogen characteristics of incident bacterial meningitis in infants in South

Africa: A Cross-sectional studyPLOS ONE

Dear Dr. Kiakuvue,

Thank you for submitting your manuscript to PLOS ONE. After careful consideration, we feel that it has merit but does not fully meet PLOS ONE’s publication criteria as it currently stands. Therefore, we invite you to submit a revised version of the manuscript that addresses the points raised during the review process.

We look forward to receiving your revised manuscript.

Kind regards,

Clinton Moodley, Ph.D.

Academic Editor

PLOS ONE

Journal Requirements:

"Bill & Melinda Gates Foundation (OPP1208882)"

5. In the online submission form, you indicated that "We can provide details of access."

**Additional Editor Comments:**

Dear Authors,  The reviewers have both indicated minor shortcomings in the original manuscript. Please address these and submit a revised manuscript for consideration. 

Reviewers' comments:

Reviewer's Responses to Questions

**Comments to the Author**

1. Is the manuscript technically sound, and do the data support the conclusions?

Reviewer #1: Partly

Reviewer #2: Partly

2. Has the statistical analysis been performed appropriately and rigorously? 

Reviewer #1: Yes

Reviewer #2: Yes

3. Have the authors made all data underlying the findings in their manuscript fully available?

Reviewer #1: No

Reviewer #2: Yes

4. Is the manuscript presented in an intelligible fashion and written in standard English?

Reviewer #1: Yes

Reviewer #2: Yes

5. Review Comments to the Author

Reviewer #1: Major comments:

• This is a nice and informative analysis of the pathogens detected in culture-positive bacterial meningitis in children aged <1 year in South Africa. The findings are very interesting, but could be made even more useful with some exploration of potential biases in the data source as well as by adding some additional information to the manuscript.

More detailed comments:

Title:

• I wouldn’t call this a cross-sectional study, as you have data over 5 years and even data on repeat cultures within the same child.

Abstract:

• Line 45: unclear what ‘and the one in the post-neonatal….’ means

Introduction:

• Line 101: could clarify that the 2.8 million cases were estimated, not actually recorded

Methods:

• Why was only data through 2018 included, instead of expanding the dataset to more recent years?

• Should mention in the methods that TB was excluded.

• Did all of the labs across the network have the same capacity for culture for the same set of pathogens?

• Does the livebirths data also come from Stats SA?

• How accurate is the census data for each year, as interpreting changes in incidence over time is heavily dependent on accurate population denominators

• Line 216: only the SIX most commonly detected pathogens were included, correct?

• Lines 224-225: do you think that infants who use public facilities differ from infants who don’t use public facilities – if so, do you think this could be biasing your incidence estimates?

• Is the 80% of the population who uses public facilities consistent across all provinces? I’m wondering if the 0.8 correction is equally applicable across every province, or if this might bias the province-specific estimates? Also, is the 80% consistent across all years 2014 through 2018?

Results

• Is it possible to include a supplemental table on which serogroups/serotypes of N. meningitidis, S. pneumoniae, H. influenzae and GBS were detected, if any of that data is available?

• Could the results of the poisson regression be presented in a supplemental table and/or the 95 CIs be added to the Figures?

Discussion:

• Lines 346-347: a potential increase in the number of samples tested is an important point. As this analysis is limited to culture-positive cases only, it’s hard to understand the surveillance performance context over time. Are any cases also tested by PCR? Did testing practices change during the study period? Did the % culture-positive change over time?

• Line 358: do you have data on which infants in this analysis were born prematurely, to better explore the disease burden/distribution in premature infants?

• Lines 361-378: I’m guessing that the pathogen distribution varies geographically, but also according to lumbar puncture practices by age group, and what pathogens labs are equipped to test for.

• Lines 395-400: do you have data on which cases received antibiotic treatment prior to lumbar puncture?

Figures

• Figures 1: There are a number of spelling errors in Figure 1

• Figure 4: are there are any notable differences in the distribution of pathogens within smaller age bands in the 28-365 day age group?

Reviewer #2: Authors should attempt to explain the reasons for the rise in incidence during the time period after 2016. The authors have summarily dwelt with above issues in discussion by mentioning in lines 344-347:

“It could be also explained by increases in transmission of bacterial infections especially in premature neonates, outbreaks in specific facilities in later years, improvements in laboratory technique and the quality of NHLS CDW data collection and data treatment, as well as an increase in the number of samples tested.”

This is one issue stands out while reading the paper, and will pique the reader’s curiosity.

The laboratory, particularly microbiology services available and its limitations/strengths could be outlined.

The authors could explore the possibility of any improvement in culture facilities of this period or was there any focussed effort to improve collection during this phase?

Did the no of csf samples generated each year change from 2014 till 2018? Was this rise because of greater sampling, better culture methods or better collection methods?

Did the number of surviving preterm neonates increase during the above period due to improvements in neonatal care? Surely, data on number of preterms delivered in the Hospitals could be available.

Did HIV positivity among the admitted patients during the period have an impact? Even though, the authors mention lack of HIV data, the HIV prevalence of the provinces could be brought out. Moreover, antenatal testing of HIV must be routine and prevalence data there could be brought out.

The percentages and figures for H. influenzae and N. meningitidis in lines 275-279 seem to be inconsistent. Please correct.

6. PLOS authors have the option to publish the peer review history of their article (what does this mean?). If published, this will include your full peer review and any attached files.

Reviewer #1: **Yes: **Heidi M. Soeters

Reviewer #2: **Yes: **Rama Krishna Sanjeev

---

## [Author Response · Author response to Decision Letter 0]

22 May 2024

All questions and comments from the editor and reviewers have been addressed and uploaded in a file titled "Response to Reviewers."

Please provide additional details regarding participant consent. In the ethics statement in the Methods and online submission information, please ensure that you have specified (1) whether consent was informed and (2) what type you obtained (for instance, written or verbal, and if verbal, how it was documented and witnessed). If your study included minors, state whether you obtained consent from parents or guardians. If the need for consent was waived by the ethics committee, please include this information.

Once you have amended this/these statement(s) in the Methods section of the manuscript, please add the same text to the “Ethics Statement” field of the submission form (via “Edit Submission”)

Response: We have now incorporated additional details and have also clarified whether all data underwent full anonymization before being accessed in the manuscript.

We note that the grant information you provided in the ‘Funding Information’ and ‘Financial Disclosure’ sections do not match. When you resubmit, please ensure that you provide the correct grant numbers for the awards you received for your study in the ‘Funding Information’ section.

Response:We have ensured that the grant numbers provided in the 'Funding Information' section matched the awards received for our study when we resubmitted.

Thank you for stating the following financial disclosure: 

"Bill & Melinda Gates Foundation (OPP1208882)"

If this statement is not correct you must amend it as needed. Please include this amended Role of Funder statement in your cover letter; we will change the online submission form on your behalf.

Response:We have added to the manuscript the correct grant number as well as the fact that the funders had no role in study design, data collection and analysis, decision to publish, or preparation of the manuscript.

In the online submission form, you indicated that "We can provide details of access."

All PLOS journals now require all data underlying the findings described in their manuscript to be freely available to other researchers, 1. In a public repository, 2. Within the manuscript itself, or 3. Uploaded as supplementary information.

This policy applies to all data except where public deposition would breach compliance with the protocol approved by your research ethics board. If your data cannot be made publicly available for ethical or legal reasons (e.g., public availability would compromise patient privacy), please explain your reasons on resubmission and your exemption request will be escalated for approval

Response:We have added that the data are available from the National Health Laboratory Service Corporate Data Warehouse (NHLS-CDW). Institutional Data Access for researchers who meet the criteria for access to confidential data.

Response:We have thoroughly reviewed our reference list to ensure its completeness and accuracy. Any references that were retracted have been updated accordingly.

Reviewer #1:

I wouldn’t call this a cross-sectional study, as you have data over 5 years and even data on repeat cultures within the same child.

Response:Our study, while spanning five years and including probable repeat cultures within the same child, remains a cross-sectional study. This classification is due to our focus on analysing data at a specific point in time or within a short timeframe, rather than tracking individuals longitudinally. We defined an episode of illness as all positive CSF samples within a 14-day period, considering samples testing positive after this period as new episodes therefore a new case. This method helps manage repeat cultures while maintaining the study's cross-sectional nature.

Line 45: unclear what ‘and the one in the post-neonatal….’ Means

Response:We have revised the text to enhance clarity.

Cause-specific Incidence risks were calculated by dividing the age-group, year of the specimen and the number of positive specimens by the appropriate mid-year population for children under 1 year old and the one in the post-neonatal period (≥ 28 days to 365 days old), while the annual numbers of registered livebirths were used for children under 28 days old.

Line 101: could clarify that the 2.8 million cases were estimated, not actually recorded

Response:We have made the modification as requested to clarify that the 2.8 million cases were estimated, not actually recorded.

Why was only data through 2018 included, instead of expanding the dataset to more recent years?

Response: The dataset included in the research only extends until 2018 because the manuscript is a transcription of a master research report, the writing of which commenced in 2019. Therefore, data beyond 2018 were not available for inclusion in the analysis at that time.

Should mention in the methods that TB was excluded.

Response: A sentence specifying that cases of TB were not included in the study has been added to the methods section as requested.

Did all of the labs across the network have the same capacity for culture for the same set of pathogens?

Response: While efforts are made to standardize procedures and protocols, disparities in resources and capabilities among laboratories across the network of public sector laboratories in South Africa are not uncommon, there may be variations in capacity for culture among different labs, particularly concerning the same set of pathogens. Factors such as infrastructure, equipment availability, staffing levels, and funding may influence the laboratory's capacity to perform culture for specific pathogens. However, the National Health Laboratory Service (NHLS) microbiology laboratories used standardised diagnostic methods.

Does the livebirths data also come from Stats SA?

Response: Yes, the livebirths data also comes from Stats SA

How accurate is the census data for each year, as interpreting changes in incidence over time is heavily dependent on accurate population denominators.

Response: The accuracy of our census data for each year is crucial for interpreting changes in incidence over time, and we took great care to ensure its reliability. All the denominator data were obtained from Statistics South Africa (STATSA) or estimated using the STATSA Sprague tool, which is widely recognized for its accuracy and reliability. STATSA is a leading authority in data collection in South Africa, known for its comprehensive and meticulous approach to gathering demographic information. Therefore, we are confident in the accuracy and validity of the population denominators used in our study.

Line 216: only the SIX most commonly detected pathogens were included, correct?

Response: Only the six most commonly detected pathogens were included for the calculation of the distribution. More information has been added in the manuscript.

Lines 224-225: do you think that infants who use public facilities differ from infants who don’t use public facilities – if so, do you think this could be biasing your incidence estimates?

Response: Yes, there might be disparities between infants using public and private healthcare facilities. However, our study used data from South Africa's largest pathology diagnostic service, which offers excellent facilities for microbiological testing and procedures and serves 80% of the total population. While this helps mitigate potential biases, we acknowledge the risk of disparity and have included this as a limitation in our study's limitations section

Is the 80% of the population who uses public facilities consistent across all provinces? I’m wondering if the 0.8 correction is equally applicable across every province, or if this might bias the province-specific estimates? Also, is the 80% consistent across all years 2014 through 2018

Response: The 80% figure represents an average rate provided by Statistics South Africa (Stats SA); a renowned institution known for its comprehensive data collection efforts. While this correction factor is widely used in our analysis to adjust for the proportion of the population utilizing public facilities, it's essential to consider potential variations across provinces and years. Stats SA employs rigorous methodologies to ensure data accuracy and reliability, but variations in healthcare infrastructure and utilization patterns may exist between provinces and over time.

Is it possible to include a supplemental table on which serogroups/serotypes of N. meningitidis, S. pneumoniae, H. influenzae and GBS were detected, if any of that data is available?

Response: Our study focused solely on the pathogen groups rather than individual serogroups/serotypes of N. meningitidis, S. pneumoniae, H. influenzae, and GBS. Therefore, we do not have data available on specific serogroups/serotypes for inclusion in a supplemental table

Could the results of the poisson regression be presented in a supplemental table and/or the 95 CIs be added to the Figures

Response: A supplemental table presenting the results of the Poisson regression analysis has been added as requested. Additionally, we have incorporated the 95% confidence intervals (CIs) to the figures for enhanced clarity and completeness of the results.

Lines 346-347: a potential increase in the number of samples tested is an important point. As this analysis is limited to culture-positive cases only, it’s hard to understand the surveillance performance context over time. Are any cases also tested by PCR? Did testing practices change during the study period? Did the % culture-positive change over time

Response; Our study specifically focused on culture samples as per our request, and we obtained the data from the NHLS CDW. Regarding your inquiries about potential changes in surveillance performance over time, it's important to note that our analysis is limited to culture-positive cases only. We did not include cases tested by PCR in our analysis. However, it's worth mentioning that there was an increase in the number of samples tested over the duration of the study, which may have contributed to the observed trends in incidence rates.

Line 358: do you have data on which infants in this analysis were born prematurely, to better explore the disease burden/distribution in premature infants?

Response:Unfortunately, we do not have data specifically identifying which infants in this analysis were born prematurely. This limitation has been noted in the study's limitations section.

Lines 361-378: I’m guessing that the pathogen distribution varies geographically, but also according to lumbar puncture practices by age group, and what pathogens labs are equipped to test for.

Response:Indeed, it's likely that the pathogen distribution varies geographically, but also according to lumbar puncture practices by age group, and what pathogens labs are equipped to test for. However, we think that the variations are less likely to be due to laboratory detection methods because the NHLS microbiology laboratories used standardised diagnostic methods and laboratories with no or limited microbiological capacity referred specimens to larger central laboratories for processing microbiology laboratories used standardised diagnostic methods.

Lines 395-400: do you have data on which cases received antibiotic treatment prior to lumbar puncture?

Response: No, we do not have data regarding which cases received antibiotic treatment prior to lumbar puncture

Figures 1: There are a number of spelling errors in Figure 1

Response: We have thoroughly reviewed Figure 1 and addressed all identified spelling errors.

Figure 4: are there are any notable differences in the distribution of pathogens within smaller age bands in the 28-365 day age group

Response: Thank you for your inquiry regarding Figure 4. Indeed, there are notable differences in the distribution of pathogens within smaller age bands in the 28-365 day age group. Specifically, in children under 28 days old in 2018, A. baumannii exhibited the highest incidence risk calculated over the study period in these two age groups. Conversely, in children aged 28 days and above (post-neonatal period), S. pneumoniae displayed the highest incidence risk, recorded at 9.8 per 100,000 persons (95% CI 7.3 – 12.7).

We have incorporated this information into our revised manuscript, and to provide further elucidation on the incidence risk across different age groups, we have included an additional table detailing incidences in various age bands. The table offers comprehensive insights into the distribution of pathogens within smaller age cohorts, enhancing the clarity and depth of our analysis.

Reviewer #2:

Authors should attempt to explain the reasons for the rise in incidence during the time period after 2016. The authors have summarily dwelt with above issues in discussion by mentioning in lines 344-347:

“It could be also explained by increases in transmission of bacterial infections especially in premature neonates, outbreaks in specific facilities in later years, improvements in laboratory technique and the quality of NHLS CDW data collection and data treatment, as well as an increase in the number of samples tested.”

This is one issue stands out while reading the paper and will pique the reader’s curiosity.

Response:We have taken note of your suggestion and have enhanced the discussion section of the paper to provide a more comprehensive explanation for the rise in incidence of bacterial meningitis cases after 2016. In addition to the factors previously mentioned, we have incorporated additional reasons, including the prevalence of antibiotic-resistant bacterial strains, changes in healthcare practices, and social determinants of health. These additions aim to provide a more nuanced understanding of the observed trend and its underlying factors.

The laboratory, particularly microbiology services available and its limitations/strengths could be outlined.

The authors could explore the possibility of any improvement in culture facilities of this period or was there any focussed effort to improve collection during this phase?

Did the no of csf samples generated each year change from 2014 till 2018? Was this rise because of greater sampling, better culture methods or better collection methods?

Response: We have now included the laboratory's strengths and limitations in our manuscript. It is important to note that from December 2014, under the leadership of a new team, many efforts were made to improve microbiology services, including culture facilities and sample collection methods at the NHLS. These enhancements likely contributed to an increase in the number of CSF samples generated each year from 2014 to 2018, reflecting greater sampling efforts, improved culture techniques, and better collect

---

## [Decision Letter · Decision Letter 1]

30 Jul 2024

PONE-D-24-04693R1Demographic and pathogen characteristics of incident bacterial meningitis in infants in South Africa: A Cross-sectional studyPLOS ONE

Dear Dr. Kiakuvue,

Thank you for submitting your manuscript to PLOS ONE. After careful consideration, we feel that it has merit but does not fully meet PLOS ONE’s publication criteria as it currently stands. Therefore, we invite you to submit a revised version of the manuscript that addresses the points raised during the review process.

We look forward to receiving your revised manuscript.

Kind regards,

Miquel Vall-llosera Camps

Senior Staff Editor

PLOS ONE

Journal Requirements:

Reviewers' comments:

Reviewer's Responses to Questions

**Comments to the Author**

1. If the authors have adequately addressed your comments raised in a previous round of review and you feel that this manuscript is now acceptable for publication, you may indicate that here to bypass the “Comments to the Author” section, enter your conflict of interest statement in the “Confidential to Editor” section, and submit your "Accept" recommendation.

Reviewer #1: (No Response)

Reviewer #2: All comments have been addressed

2. Is the manuscript technically sound, and do the data support the conclusions?

Reviewer #1: Yes

Reviewer #2: Yes

3. Has the statistical analysis been performed appropriately and rigorously? 

Reviewer #1: Yes

Reviewer #2: Yes

4. Have the authors made all data underlying the findings in their manuscript fully available?

Reviewer #1: Yes

Reviewer #2: Yes

5. Is the manuscript presented in an intelligible fashion and written in standard English?

Reviewer #1: Yes

Reviewer #2: Yes

6. Review Comments to the Author

**Reviewer #1:** Major comments:

• This is a nice and informative analysis of the pathogens detected in culture-positive bacterial meningitis in children aged <1 year in South Africa. The findings are very interesting, and the revisions have improved the clarity and utility of the manuscript. Only a few minor considerations for improvement remain.

Title:

• I respectfully disagree with this study being classified as a cross-sectional study. Incidence calculations are reported in this manuscript, and incidence cannot be calculated from cross-sectional studies.

Abstract:

• Line 49: would be helpful to specify: We identified 3575 (1.5) CASES of culture-confirmed bacterial meningitis…

Introduction:

• Line 122: should say ‘a quadrivalent protein-polysaccharide meningococcal VACCINE…’

Methods:

• In the Source of Data section, should mention that the livebirths data also come from Stats SA

• Line 207: only the SIX most commonly detected pathogens were included in the distribution calculations, correct? The text in this one sentence currently says seven.

Results

• Lines 290-292: it is unclear what the denominator is for these reported proportions for H. influenzae and N. meningitidis.

**Reviewer #2:** The authors have done due diligence in improving the manuscript. They have answered my comments satisfactorily.

7. PLOS authors have the option to publish the peer review history of their article (what does this mean?). If published, this will include your full peer review and any attached files.

Reviewer #1: **Yes: **Heidi M Soeters

Reviewer #2: **Yes: **Rama Krishna Sanjeev

---

## [Author Response · Author response to Decision Letter 1]

3 Aug 2024

Thank you for your detailed feedback and suggestions on our manuscript. Below, we address each of your comments and outline the changes made accordingly:

Study Design:

Original Comment: "I respectfully disagree with this study being classified as a cross-sectional study. Incidence calculations are reported in this manuscript, and incidence cannot be calculated from cross-sectional studies."

Response: As requested, the study design has been updated and is now described as a retrospective cohort study, which is more appropriate for the methods and analyses conducted.

Original Comment: "would be helpful to specify: We identified 3575 (1.5) CASES of culture-confirmed bacterial meningitis…"

Response: This has been corrected as requested.

Meningococcal Vaccine:

Original Comment: "Line 122: should say ‘a quadrivalent protein-polysaccharide meningococcal VACCINE"

Response: The text has been revised to state ‘a quadrivalent protein-polysaccharide meningococcal vaccine’ as requested.

Source of Data:

Original Comment: "In the Source of Data section, should mention that the livebirths data also come from Stats SA."

Response: This has been done. We have now mentioned that the live births data also come from Stats SA.

Pathogen Distribution Calculations:

Original Comment: "Line 207: only the SIX most commonly detected pathogens were included in the distribution calculations, correct? The text in this one sentence currently says seven."

Response: This has been corrected to accurately reflect that only the six most commonly detected pathogens were included in the distribution calculations.

Proportions Denominator:

Original Comment: "Lines 290-292: it is unclear what the denominator is for these reported proportions for H. influenzae and N. meningitidis."

Response: More clear sentences have been added to the manuscript to clarify the denominator used for these reported proportions.

We hope these revisions address your concerns effectively. Thank you for your valuable feedback, which has greatly improved the clarity and accuracy of our manuscript.

---

## [Decision Letter · Decision Letter 2]

4 Sep 2024

Demographic and pathogen characteristics of

incident bacterial meningitis in infants in South

Africa: A cohort study

PONE-D-24-04693R2

Dear Dr. Kiakuvue,

We’re pleased to inform you that your manuscript has been judged scientifically suitable for publication and will be formally accepted for publication once it meets all outstanding technical requirements.

Kind regards,

D. William Cameron, MD

Academic Editor

PLOS ONE

Additional Editor Comments (optional):

Reviewers' comments:

Reviewer's Responses to Questions

**Comments to the Author**

1. If the authors have adequately addressed your comments raised in a previous round of review and you feel that this manuscript is now acceptable for publication, you may indicate that here to bypass the “Comments to the Author” section, enter your conflict of interest statement in the “Confidential to Editor” section, and submit your "Accept" recommendation.

Reviewer #1: All comments have been addressed

Reviewer #2: All comments have been addressed

2. Is the manuscript technically sound, and do the data support the conclusions?

Reviewer #1: Yes

Reviewer #2: Yes

3. Has the statistical analysis been performed appropriately and rigorously? 

Reviewer #1: Yes

Reviewer #2: Yes

4. Have the authors made all data underlying the findings in their manuscript fully available?

Reviewer #1: Yes

Reviewer #2: Yes

5. Is the manuscript presented in an intelligible fashion and written in standard English?

Reviewer #1: Yes

Reviewer #2: Yes

6. Review Comments to the Author

Reviewer #1: (No Response)

Reviewer #2: The authors have addressed my concerns in the present iteration. The work adequately addresses the research question.

7. PLOS authors have the option to publish the peer review history of their article (what does this mean?). If published, this will include your full peer review and any attached files.

Reviewer #1: **Yes: **Heidi M. Soeters

Reviewer #2: **Yes: **Rama Krishna Sanjeev

---

## [Editor Report · Acceptance letter]

15 Sep 2024

PONE-D-24-04693R2 

PLOS ONE

Dear Dr. Kiakuvue, 

I'm pleased to inform you that your manuscript has been deemed suitable for publication in PLOS ONE. Congratulations! Your manuscript is now being handed over to our production team.

Kind regards, 

on behalf of

Professor D. William Cameron 

Academic Editor

PLOS ONE